# A Novel Preanalytical Strategy Enabling Application of a Colorimetric Nanoaptasensor for On-Site Detection of AFB1 in Cattle Feed

**DOI:** 10.3390/s22239280

**Published:** 2022-11-29

**Authors:** Braulio Contreras-Trigo, Víctor Díaz-García, Patricio Oyarzún

**Affiliations:** Facultad de Ingeniería, Arquitectura y Diseño, Universidad San Sebastián, Lientur 1457, Concepción 4080871, Chile

**Keywords:** aflatoxin B1, cattle feed, colorimetric aptasensor, AuNPs, biosensors, aptamer

## Abstract

Aflatoxin contamination of cattle feed is responsible for serious adverse effects on animal and human health. A number of approaches have been reported to determine aflatoxin B1 (AFB1) in a variety of feed samples using aptasensors. However, rapid analysis of AFB1 in these matrices remains to be addressed in light of the complexity of the preanalytical process. Herein we describe an optimization on the preanalytical stage to minimize the sample processing steps required to perform semi-quantitative colorimetric detection of AFB1 in cattle feed using a gold nanoparticle-based aptasensor (nano-aptasensor). The optical behavior of the nano-aptasensor was characterized in different organics solvents, with acetonitrile showing the least interference on the activity of the nan-aptasensor. This solvent was selected as the extractant agent for AFB1-containing feed, allowing for the first time, direct colorimetric detection from the crude extract (detection limit of 5 µg/kg). Overall, these results lend support to the application of this technology for the on-site detection of AFB1 in the dairy sector.

## 1. Introduction

Aflatoxins are secondary metabolites produced by fungal species of the genus Aspergillus and other toxigenic strains of molds [1]. These compounds are associated with significant economic losses in agriculture, crop farming, and livestock activities [2,3], having an impact on 25% of the world’s crop harvest [4,5]. In addition, aflatoxins represent a serious hazard to food safety because of their worldwide distribution and high carcinogenicity, among which aflatoxin B1 (AFB1) is the most harmful type and responsible for the greatest economic and health burden worldwide [6]. Overall, about 28% of the total worldwide cases of hepatocellular carcinoma (up to 172,000 cases per year) are caused by aflatoxin exposure [7].

AFB1 is classified by the International Organization for Research in Cancer (IARC) as a class 1 carcinogen (agents that are certainly carcinogenic for humans, with epidemiological evidence as the causative agent of human hepatocellular carcinomas (liver cancer) [8]. AFB1 can contaminate numerous food and feed products, such as cereals, oilseeds, spices, and tree nuts, during harvesting and storage, and due to poor processing conditions [9]. Likewise, cattle can be exposed to aflatoxins through corn grain, corn silage, and corn processing by-products. AFB1 is hepatotoxic and hepatocarcinogen in dairy cattle, increasing the susceptibility to infectious diseases and reducing efficiency in milk production and feed intake [10]. Upon ingestion via contaminated feeds, AFB1 can be partially metabolized by cows into aflatoxin M1 (AFM1), which is a hydroxylated derivate classified by the IARC in group 2B (possibly carcinogenic to humans). Thus, AFB1 and AFM1 can be both secreted through the milk of dairy cows and have the potential to reach the human food chain [8,11,12].

Rapid detection of AFB1 in cattle feed is, therefore, paramount to avoid its negative effects on animal health and to prevent its presence in milk, which causes economic losses for livestock farmers and impacts public health [13,14]. However, the feed is a complex matrix that possesses several preanalytical and analytical challenges. This matrix often consists of ingredients such as hay, soybean, wet-grain corn, molasses, which are typically present in TMR feed (total mixed ratio) [15]. These ingredients have the potential to generate interference in analytical procedures [16]. Feed samples are typically subjected to an extraction step with polar organic solvents to recover the aflatoxin from the matrix, followed by a purification step to remove the extractant and concentrate the analyte in a lower volume of aqueous solution [5]. Accordingly, the detection and quantification of aflatoxins in the feed are dependent on efficient extraction steps using aqueous–organic solvent mixtures [17]. Methanol, acetonitrile, or acetone mixed in different proportions with small amounts of water are often employed because of the low water solubility of AFB1 [18,19,20,21].

The maximum residue levels (MRLs) accepted for AFB1 in the feed of dairy cows fluctuate from 5 µg/kg (European Union) [22] to 20 µg/kg (United States) [23] and 20 µg/kg (China) [24]. Conventional analytical approaches for aflatoxin identification include thin-layer chromatography (TLC), high-performance liquid chromatography (HPLC), and gas chromatography (GC). However, immunoassays are increasingly becoming available for point-of-care testing due to advantages such as ease of use, high specificity, and cost-effectiveness [25]. Enzyme-linked immunosorbent assay (ELISA) has been well-studied for AFB1 detection in feedstuffs [26,27] and commercial kits have recently proved to be competitive with HPLC [28]. Likewise, novel immunosensors are under active development to improve analytical variables and to enable the on-site application of the devices [29,30,31,32,33]. However, despite these signs of progress, several challenges remain to be addressed to avoid under or overestimation of AFB1 concentration in the samples, including factors such as the preanalytical procedures, the composition of the extraction solvent, co-extracted compounds (matrix interference), and cross-reactivity of antibodies with molecules similar to the target [34]. Of particular relevance is the instability of antibodies in the organic solvents that are commonly used for AFB1 extraction, such as methanol/water mixtures [35]. The crude extract is typically passed through immunoaffinity columns and further washing steps to remove impurities [36], as a requirement to analyze AFB1 by ELISA or other immunoassays.

DNA aptamers are a promising class of bioreceptors consisting of short oligonucleotides that bind with high specificity to non-nucleotide analytes [37]. In comparison to antibodies, aptamers have superior stability and can be produced for a wide range of low molecular weight, toxic, and nonimmunogenic compounds [38]. Thus, they offer huge advantages over protein-based antibodies due to their higher stability and their capability to recover their native conformation after denaturation by heat treatment [39,40]. The combination of gold nanoparticles (AuNPs) with aptamers (NAS; nano-aptasensor) is gaining considerable attention to investigate the detection of a variety of analytes in the field of food safety [41]. AuNPs allow the development of label-free aptasensors due to their surface plasmon resonance (SPR) properties, which result from the collective oscillation of the conduction electrons in the presence of incident light and causes a sharp and intense absorption band at 520 nm [42]. AuNP-based NASs enable colorimetric detection by following a red-to-purple blue shift of the absorption spectrum during the aggregation of the nanoparticles, providing a versatile, sensitive, selective, and easy-to-apply platform. We recently reported NAS systems for the colorimetric detection of antibiotic residues from raw milk samples, addressing both the preanalytical problem [39] and the application of a machine-learning algorithm to improve the sensitivity of the analytical process [43]. Aptasensors for AFB1 detection have been successfully reported in rice, peanuts, and wheat flour upon extraction using methanol/water in 80/20 and 60/40 mixtures [19,44]. However, these methods require at least one step to remove the organic extractant prior to detection [19,45]. Thus, ELISA-based techniques perform sample cleanup by passing the crude extract through immunoaffinity columns and further washing to remove impurities [36].

In food analysis, the complexity of the matrices can have the largest impact on the performance of the methods [46,47]. Different types of NAS-based assays for AFB1 detection have been reported, performing detection for peanuts rice, flour, maize meal, and corn [48,49,50]. The low amount of AFB1 in the samples (trace-level) combined with the complexity of the matrix are common characteristics of these studies, which include preanalytical operations allowing elimination of the extraction solvent followed by concentration of AFB1 in aqueous buffers. While these nanobiosensors are operational in the laboratory, the transition to real-world applications requires several obstacles to be overcome [51]. Food and agricultural applications can be especially difficult due to the breadth of chemical environments and potential interferents causing alterations to the analytical procedures and/or preventing access to the analytes in the matrices [52].

Therefore, we aimed to develop a preanalytical strategy that provides an improvement toward enabling the in-field application of nanobiosensors, in line with recent literature addressing the challenges this technology faces on the road to market [53]. To the best of our knowledge, this is the first time a methodology is specifically reported to enable the detection of AFB1 from cattle feed directly in the crude extract. The extraction of AFB1 and the optical behavior of the NAS were investigated using different organics solvents, with the aim of minimizing the steps needed to implement the colorimetric assay [17]. The method developed herein offers a combination of simplicity and sensitivity, holding the potential to contribute to the development of NAS technology with applications for on-site detection of AFB1 in dairy farms.

## 2. Materials and Methods

### 2.1. Reagents and Chemicals

Tetrachloroauric acid (HAuCl_4_·3H_2_O), Sephadex G-25, AFB1 standard, trisodium citrate, sodium chloride, acetonitrile, ethanol, isopropanol, and methanol were purchased from Merck (Darmstadt, Germany). Dithiothreitol and phosphate buffer were supplied by Winkler (Santiago, Chile). The single-stranded DNA aptamer was synthesized by Integrated DNA Technologies Inc. (IDT, Coralville, IA, USA).

### 2.2. NAS and Experimental Strategy

The preanalytical strategy was investigated with the aim to simplify the sample processing steps enabling NAS-based detection of AFB1 in cattle feed. The NAS was synthesized by conjugating a previously reported AFB1-specific ss DNA aptamer to the AuNPs surface [19], which adopts a flexible random coil linear structure that allows the aptamer bases to interact with the nanoparticle surface through electrostatic forces [54]. Figure 1 provides a schematic description of the detection reaction, which shows the aptamers coating the AuNPs surface and inhibiting salt-induced aggregation due to electrostatic repulsions among the nanoparticles. However, in the presence of AFB1, the aptamers adopt a folded structure that leads to a decrease in surface protection and subsequent aggregation of the AuNPs upon NaCl addition. Aggregation is proportional to the AFB1 concentration and can be followed spectrophotometrically.

AuNPs:aptamer molar ratios of 1:10, 1:20, and 1:40 were first investigated to determine the best parameters and conditions for maximizing in acetonitrile the colorimetric signal of the NAS for AFB1 detection, on the basis that AFB1 is highly soluble in acetonitrile and this solvent was almost unexplored in the literature for this application. Subsequently, acetonitrile was compared with other organic solvents that are typically used as extractant agents of AFB1 (methanol, ethanol, and isopropanol) in terms of their effects on the process of salt-induced NAS aggregation. The aim of this experiment was to investigate if acetonitrile has properties that are advantageous to preserve NAS activity in comparison with the three alcohols, thus, selecting an organic medium with the least interference on the optical signal of the NAS. Finally, detection assays of AFB1 were carried out directly in the crude extract to prove the capability of acetonitrile to recover AFB1 from cattle feed and to determine its concentration with a sensitivity able to meet international standards of MRL compliance in cattle feed.

### 2.3. NAS Preparation

AuNPs were synthesized through the citrate reduction method [55,56] after adjusting the pH to 5.3 [57]. Aptamer-functionalized AuNPs were prepared by functionalization of the nanoparticles based on the protocol described by Hill and Mikrin (2006) [58]. Briefly, 5 nmol of the thiolated aptamer specific for AFB1 (GCA CTA CTC CCT AAC ATC TCA AGC GTT GGG CAC GTG TGT CTC TCT GTG TCT CGT GCC CTT CGC TAG GCCC/3ThioMC3-D) [19] were reduced by incubation with dithiothreitol 0.1 M in phosphate buffer (pH 8) for 3 h, at room temperature and in darkness. 3′-SH-aptamers were subsequently purified by gel filtration with Sephadex G-25 and their concentration was determined at 260 nm with an Epoch^TM^ Microplate Spectrophotometer (Bio-Tek Instruments, Winooski, VT, USA). Finally, AuNPs solutions were incubated with the aptamer in molar ratios of 1:10, 1:20, and 1:40 (AuNP:aptamer), stirring the mixtures at 1200 rpm for two days in the dark (20 °C).

### 2.4. NAS Characterization

Particle size and morphology characterization were carried out by transmission electron microscopy (TEM) with a 4 Å resolution (TEM; JEOL-JEM 1200EX-II, Tokyo, Japan) using a Gatan CCD camera for image acquisition (model 782; Gatan, Inc., Pleasanton, CA, USA). The surface charge of the NAS was determined by dynamic light scattering (DLS) using a Zetasizer Nano-ZS90 analyzer (Malvern Instruments, Westborough, MA, USA). The concentration of AuNPs was calculated spectrophotometrically at 520 nm (Ɛ = 2.01 × 10^8^ M^−1^ cm^−1^) with an Epoch^TM^ Microplate Spectrophotometer (Biotek Instruments, Winooski, VT, USA) [59]. The optical behavior of the NAS was evaluated in presence of organic solvent (acetonitrile, ethanol, isopropanol, and methanol) and compared with water to investigate the effect of the solvents on nanoparticle aggregation. The colorimetric response of the NAS was determined at different concentrations of AFB1 (standard solution) after heating the NAS at 95 °C for 5 min to linearize the aptamer on the nanoparticle surface.

### 2.5. AFB1 Extraction

Cattle feed provided by a dairy farm was dried at 40 °C for 15 h and then fine-grained with a blade mill. Pretreated samples (2 g) were spiked with 200 µL of AFB1 standard solution (in acetonitrile) to generate AFB1-spiked samples at concentrations of 1, 2, 5, and 10 µg/kg (in triplicate). Spiked samples (2 g) were treated with 8 mL of acetonitrile and the resulting mixture was stirred at 2500 rpm for 20 min, at 20 °C, and then centrifuged at 3500 rpm for 10 min, at 4 °C. Finally, the supernatant was filtered with a 0.2 µm PVDF syringe filter and stored in the dark at 4 °C for further analysis. AFB1 concentration in the samples is expressed as the amount of toxin by mass of dry powdered feed.

### 2.6. AFB1 Detection Assay

A typical assay in microplate wells consisted of 200 µL of sample incubated with 100 µL of the activated NAS (13 nM final) at 25 °C for 10 min, and subsequently cooled down at room temperature. Then, 60 µL of NaCl 1 M (0.17 M final) was added into the solutions and incubated for 10 min at 25 °C to monitor the aggregation process. AuNP aggregation data were analyzed spectrophotometrically by following the shift of the plasmon resonance peak from 520 nm to 620 nm (A_520_/A_620_) and 700 nm (A_520_/A_700_). The measurements for AFB1 detection were expressed as the magnitude in the fold changes of the colorimetric signal produced in AFB1-containing crude extract (treatments) in comparison to samples without AFB1 (controls), according to Equation (1). NAS absorbances were also corrected by subtracting the values against nanopure water (blank).
(1)$$\mathrm{Fold}\;\mathrm{OD}\;620\;\mathrm{nm}=\frac{620\;\mathrm{nm}\;\mathrm{Signal}\;\mathrm{of}\;\mathrm{NAS}\;\mathrm{with}\;\mathrm{AFB}1\;\left(\mathrm{treatment}\right)}{620\;\mathrm{nm}\;\mathrm{Signal}\;\mathrm{of}\;\mathrm{NAS}\;\mathrm{without}\;\mathrm{AFB}1\;\left(\mathrm{control}\right)\;}$$

### 2.7. Statistical Analysis

Data shown are the mean ± standard error of at least three independent experiments. Statistical significance was determined at the 95% confidence level, using the non-parametric Mann–Whitney test to compare differences between two groups, and the non-parametric multiple ANOVA test for multiple comparisons. The statistical significance of the slopes was calculated by Pearson correlation (*p* < 0.05) using linear regression analysis.

## 3. Results

### 3.1. NAS Characterization

The obtained AuNPs and NAS presented a spherical shape and average diameters of ~20 nm (Figure 2A,B). In addition, their identical optical behaviors demonstrated aptamer conjugation without changes to the nanoparticle structure (Figure 2C). The salt-induced aggregation and zeta potential (pZ) showed differences due to negative charges provided by the aptamers on the NAS surface (Figure 2C,D). This is reflected by a more negative value of pZ for the NAS (−35 mV) in comparison with the AuNPs (−26 mV) and variations in the SPR peaks (520/620 nm) during AuNPs aggregation.

### 3.2. Determination of the Detection Parameters of NAS

AuNP:aptamer molar ratios of 1:10, 1:20, and 1:40 were investigated to select conditions that maximize the colorimetric signal generated by the NAS in acetonitrile. Each NAS was characterized by the detection of AFB1 at 0, 0.25, 0.5, 1, and 2 times the concentration corresponding to the MRL in the European Union [22]. The OD ratio 620/520 nm is widely used to follow the aggregation of colorimetric sensors based on AuNPs [60,61], even though we showed in a previous work that this parameter is not necessarily the best way to perform analyte detection with NAS [39]. With this in mind, NAS aggregation was analyzed spectrophotometrically between 400 and 750 nm to select the reading parameters that maximize the signal associated with AFB1 detection. Figure 3 presents the colorimetric response (fold changes; Equation (1)) for each NAS upon the addition of NaCl (see spectra in Appendix A).

The strongest signal to monitor AFBI detection (at MRL concentration level) was obtained with 1:10 and 1:20 AuNP:aptamer molar ratios when reading the absorbance at 520, 620, and 720 nm (Figure 3A), instead of using an absorption relationship at two different wavelengths. The 1:10 molar ratio (NAS) showed the highest discrimination capability of AFB1 at concentrations ranging from 0 to 10 µg/L (Figure 3B). A multiple comparison analysis showed that the increase in the optical signal is statistically significant at AFB1 concentrations between 0 and 5 µg/L (Figure 3C), showing an excellent linear correlation in this range (R^2^ = 0.99) (Figure 3D). In addition, Table 1 shows the slope for this NAS is significantly different from zero (*p*-value: 0.0326) and presents the resulting values for the detection limit (LOD) and quantification limit (LOQ) determined by linear regression analysis.

### 3.3. Effect of Organic Solvents on Salt-Induced NAS Aggregation

AFB1 detection assays were previously carried out with acetonitrile, despite the fact that the use of alcohols for AFB1 extraction is typically indicated in the literature [62]. To better understand the differences between solvents, the aggregation behavior of the NAS was characterized in acetonitrile, methanol, ethanol, and isopropanol by following the absorption spectra upon the addition of 1 M NaCl (0.17 M final). Figure 4A shows spectra illustrating the red to blue–purple shifts for each polar organic solvent in comparison with water. Acetonitrile was the organic media with the lowest effect on the optical behavior of the NAS, showing no statistical difference in comparison to water upon salt-induced aggregation (Figure 4B). This result provides strong evidence in favor of acetonitrile as an organic medium to enable AFB1 detection, with the enormous benefit of avoiding the need to remove the extractant from the crude extract before the analytical phase.

### 3.4. AFB1 Detection Assay from Spiked Samples

Figure 5A shows the absorption spectra of the NAS in the acetonitrile-based crude extract. Absorption peaks at 415 and 660 nm are probably attributable to the presence of chlorophyll in the feed [63]; however, the presence in the extract of other molecules of vegetal origin cannot be ruled out [64]. AFB1 detection under these conditions was successfully followed from the spectral shift, by measuring the OD at 620 nm in the concentration range 0–10 μg/kg. Ultimately, the NAS was capable to discriminate AFB1 at a minimum concentration of 5 μg/kg (Figure 5B).

Figure 6 summarizes the extraction process of AFB1-spiked feed samples using acetonitrile as the extractant agent, followed by direct spectrophotometric detection of the analyte in the crude extract. Briefly, dried and fine-powdered feed samples were spiked with AFB1 standard solution and employed for acetonitrile-based AFB1 extraction (preanalytical phase). An aliquot of the crude extract was then used for direct analysis with the NAS, proving that semi-quantitative detection of AFB1 can be achieved with a detection limit of 5 µg/kg (analytical phase). The total time required for the analysis was approximately 1 h.

## 4. Discussion

This research focused on optimizing a preanalytical strategy to enable AFB1 assay, given that this molecule is among the most prevalent and carcinogenic mycotoxins found in feedstuffs [65]. Importantly, the aptamer employed herein was previously studied by Chen et al., who proved a strong specificity for AFB1 (at 100 nM) in the presence of most common mycotoxins: ochratoxin A (OCA), aflatoxin M1 (AFM1), aflatoxin G1 (AFG1), aflatoxin G2 (AFG2), aflatoxin B2 (AFB2), zearalenone (ZEN), deoxynivalenol (DON), and fumonisins (FB1). These mycotoxins were unable, at a concentration of 1 uM, to induce a response in the aptasensor developed by the authors [19]. The co-contamination of agricultural products with multiple mycotoxins is frequently observed [66,67]. However, as the normal concentration range of mycotoxins in feed is in the nanomolar range [68], we could reasonably expect that the NAS will not be affected by the presence of other mycotoxins.

Our results prove acetonitrile is capable of outperforming alcohols from the point of view of the optical behavior of the NAS, while enabling direct detection of AFB1 from the crude extract (Figure 2 and Figure 5), in comparison with previous studies performing AFB1 extraction from feed samples with pure methanol [69] and mixtures of methanol–water mixtures [19], or acetonitrile–water [70]. The effects of acetonitrile on NAS recognition are complex and have not been specifically studied. On the one hand, organic solvents have the potential to alter the three-dimensional structure of the aptamers that is required to bind the analyte. For example, low concentrations of alcohols (methanol, ethanol, and 2-propanol) were shown to inhibit the interaction of an anti-D-adenosine aptamer with its target molecule [71]. However, antagonistic effects can be induced over the stability of the NAS, as well as over the strength of the aptamer-analyte interaction. We hypothesize that acetonitrile has the potential to increase the strength of aptamer-AFB1 interactions by excluding water molecules and, thus, increasing the number of hydrogen-bonding interactions between the aptamer and its target. For example, a study addressing the hydrogen-bonding interactions of DNA with 2-Imidazolidinethione showed that a higher fraction of acetonitrile in aqueous DNA solution inhibited the competition of water molecules (through hydrogen bounds) and led to an increase in the strength of the interaction between the DNA and the target analyte [72]. These interactions are mainly associated with hydrogen bridges, but aromatic stacking and van der Waals forces through nitrogen bases can make an appreciable contribution [73]. However, an effect in the opposite direction can be expected by disruption of the electrostatic interactions between the aptamer and the AuNP surface [74]. Appendix A demonstrates this effect, since acetonitrile induced aggregation of non-functionalized AuNPs in the absence of NaCl, by interfering in the repulsive layer that stabilizes the NAS [75]. Accordingly, aptamers must be covalently linked to the AuNP surface in order to keep the functionality of the NAS in an acetonitrile-containing reaction medium.

In 2015, several works were published that addressed the employment of polar organic solvents for AFB1 extraction from feed, followed by sample purification and detection with NAS (Table 2). Despite these methodologies offering strong analytical performance (LODs at ppt level), the current work is the first time acetonitrile is reported as the extractant agent to recover AFB1 from a complex feed matrix without additional operations enabling detection. Importantly, our procedure does not require steps such as dilution of the crude extract in aqueous solution and purification/washing operations to remove the solvent and/or concentration of the analyte, either by column elution or by evaporation. In addition, by spiking AFB1 directly into the feed samples we provide a more realistic approach to the contamination problem than other works performing this process after the extraction step.

Herein, a LOD value of 0.88 ppb for the NAS was calculated from the calibration curve with the AFB1 standard solution (in pure acetonitrile). However, this parameter increases to 5 ppb for simulated applications using AFB1-containing feed samples and acetonitrile extraction. This difference is likely related to the loss of AFB1 during the extraction process. Importantly, this new approach makes a significant contribution to simplifying the experimental procedure and shortening the time needed to achieve a competitive LOD value. Even though it has a lesser sensitivity than other methodologies based on similar principles (Table 2), it meets the most stringent international standard of MRL compliance in cattle feed (5 μg/kg in the EU). Overall, this method holds the potential to be further developed as a point-of-care assay with application in the dairy sector [79].

## 5. Conclusions

The optimization on the preanalytical stage was successfully implemented to minimize the sample processing steps required to perform colorimetric detection of AFB1 from cattle feed. Acetonitrile outperformed typically reported organic media (alcohol-water mixtures) by inducing a negligible effect on the optical signal of the NAS under detection conditions (at 600 nm). Importantly, this solvent allowed direct colorimetric detection in the crude extract with the NAS, with no need for further purification and/or concentration steps of the analyte. Thus, this method offers a combination of simplicity and sensitivity, proving able to discriminate the presence of AFB1 in the feed at concentration levels compatible with international regulatory limits (5 μg/kg). The proposed experimental strategy holds the potential to contribute to the development of nano-aptasensor technology with applications for the on-site detection of AFB1 in the dairy sector.

## Figures and Tables

**Figure 1 sensors-22-09280-f001:**
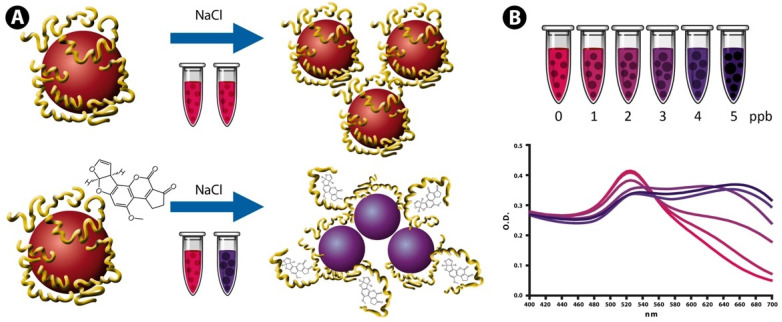
Schematic illustration showing the sensing principle of aptamer-conjugated AuNPs for colorimetric detection of AFB1 present in feed. (**A**) Covalently conjugated aptamers in a randomly coiled linear structure inhibit salt-induced aggregation, while the conformational change induced by AFB1 interaction decreases surface protection and allows AuNP aggregation upon NaCl addition. (**B**) Absorption spectra of the AuNPs showing antibiotic-induced aggregation of the nanoparticles and the resulting shift in the plasmon resonance peak (from red to blue-purple).

**Figure 2 sensors-22-09280-f002:**
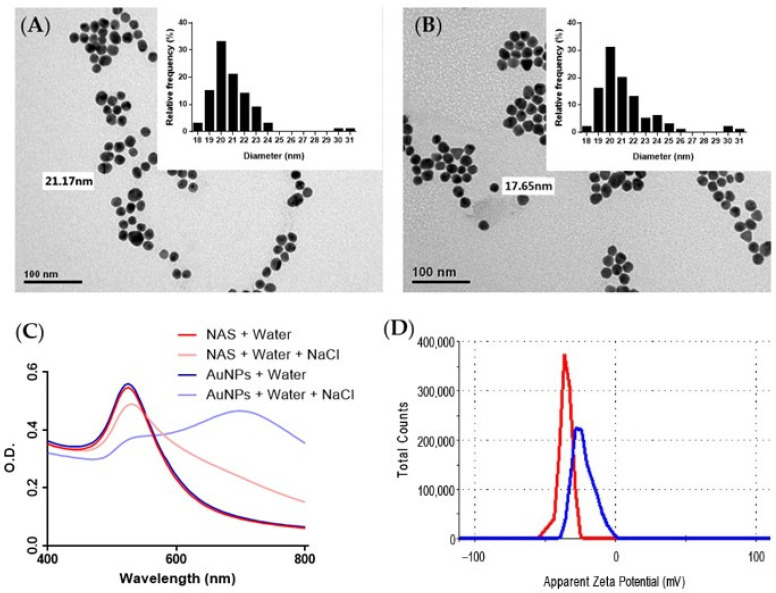
TEM micrographs of (**A**) AuNPs and (**B**) NAS, showing the average diameters and particle size distribution (histograms insets). Characterization of the nanoparticles is presented in terms of (**C**) zeta potential of AuNPs (blue line) and NAS (red line), and (**D**) absorption spectra in the visible range for AuNPs and NAS upon NaCl addition.

**Figure 3 sensors-22-09280-f003:**
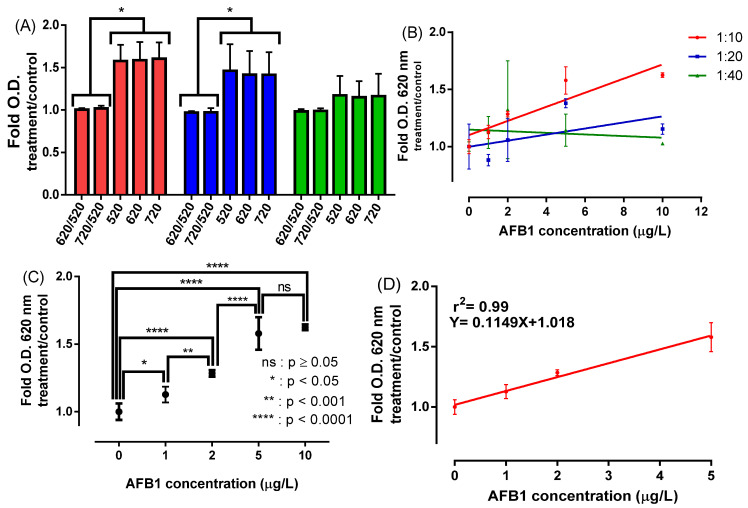
Detection assays of AFB1 in acetonitrile. (**A**) Colorimetric response of the NAS using different wavelengths in AFB1-containing samples (at MRL concentration) and controls of AFB1-free samples. (**B**) Colorimetric response using AuNPs:aptamer molar ratios of 1: 10 (red), 1:20 (blue), and 1:40 (green) for detection of AFB1 at concentrations of 0, 1, 2, 5, and 10 ug/L. (**C**) Statistical analysis of the colorimetric response of NAS 1:10 molar ratio as a function of AFB1 concentration by using the Kruskal–Wallis test; (**D**) Linear fitting of the colorimetric response data as a function of the AFB1 concentration of NAS 1:10 molar ratio. Asterisks denote statistically significant differences between the treatments and controls. * *p* < 0.05, ** *p* < 0.001 and **** *p* < 0.0001. Non-significant differences are indicated as ns.

**Figure 4 sensors-22-09280-f004:**
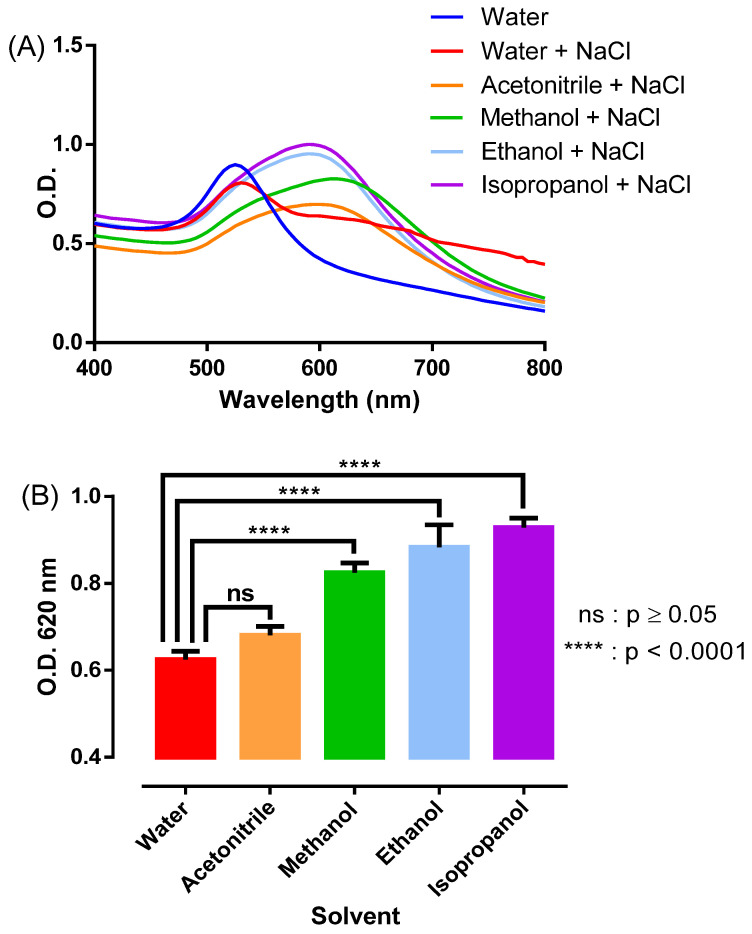
Comparative analysis of salt-induced NAS aggregation in water, acetonitrile, methanol, ethanol, and isopropanol. (**A**) Absorption spectra resulting from incubating the NAS for 30 min in each organic solvent; (**B**) Differences in absorbance at 620 nm between water and the organic solvents. The results were averaged from three independent experiments (n = 3). Asterisks denote statistically significant differences between the water and organic solvents. **** = *p* < 0.0001. Non-significant differences are represented as ns.

**Figure 5 sensors-22-09280-f005:**
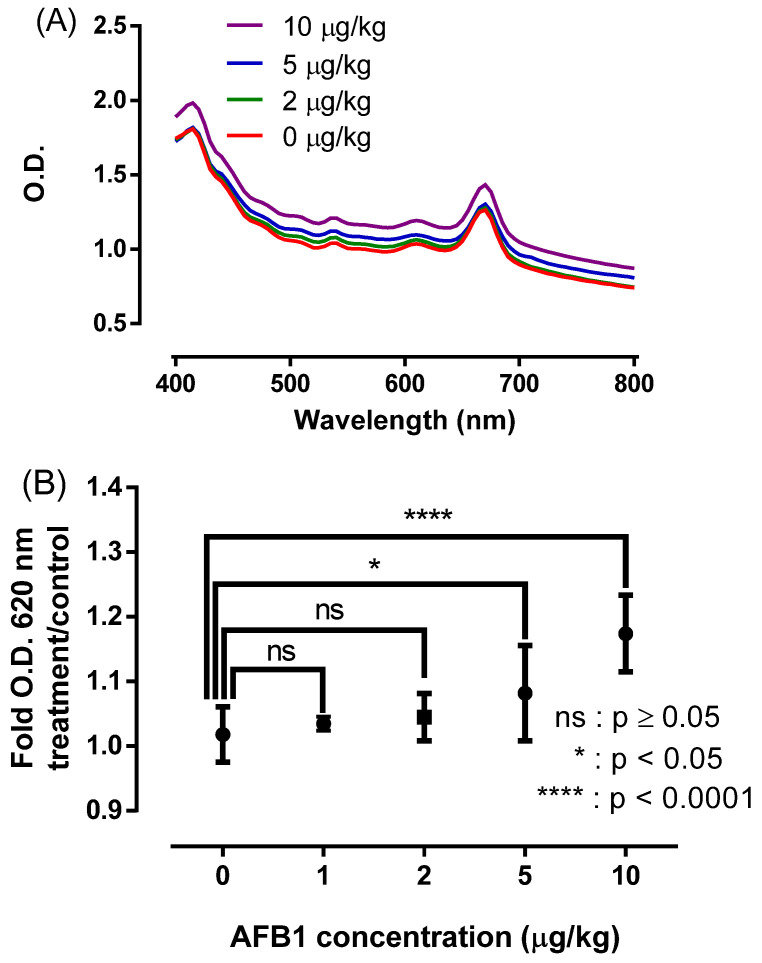
NAS-based detection assay of AFB1 present in cattle feed using acetonitrile as the extractant agent, showing (**A**) the absorption spectra in the crude extract and (**B**) AFB1 detection using O.D. at 620 nm. The results were averaged from three independent experiments (n = 3). Statistically significant differences compared with the controls and different treatments are indicated. * *p* < 0.05 and **** *p* < 0.0001.

**Figure 6 sensors-22-09280-f006:**
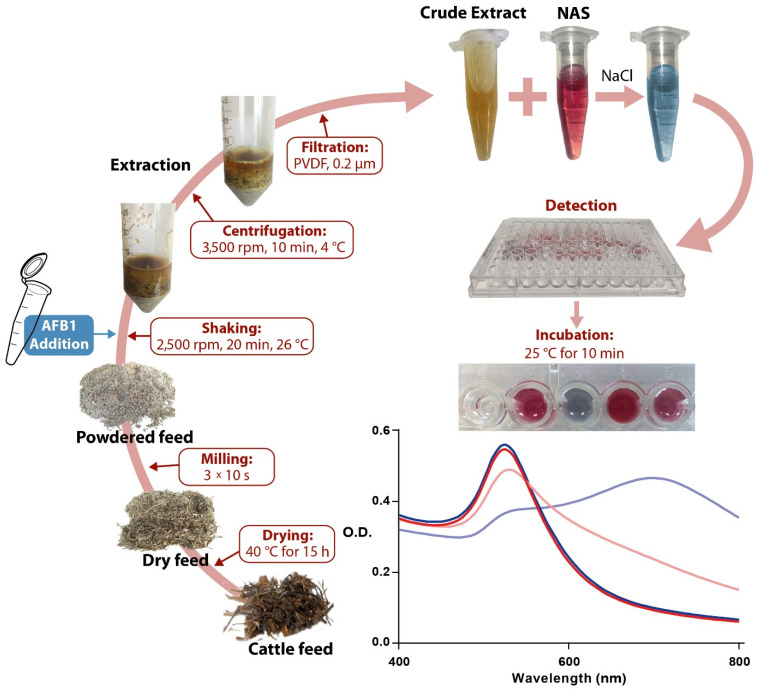
Schematic illustration of preanalytical and analytical phases of the NAS assay to detect AFB1 in cattle feed samples.

**Table 1 sensors-22-09280-t001:** Sensibility parameters for AFB1 detection with NAS using different AuNP:aptamer ratios.

NAS	Equation of the Line	R^2^	LOD (µg/L)	LOQ (µg/L)	Slope
1:10	Y = 0.06171 × X + 1.101	0.826	0.88	2.94	*p*-value = 0.0326
1:20	Y = 0.02658 × X + 0.9989	0.328	9.50	31.67	*p*-value = 0.3131
1:30	Y = −0.007129 × X + 1.149	0.051	-	-	*p*-value = 0.7145

**Table 2 sensors-22-09280-t002:** AFB1 detection assays for feed samples based on different types of nano-aptasensors.

Organic Extractant	Feed	AFB1 Sample Spiking	Final Sample Medium	NP	Detection Principle	LOD	Concentration Factor	Ref.
Methanol-water (8:2)	Rice	Before	Aqueous Buffer	AuNPs	Colorimetric	2 pM (0.0006 ppb)	20	[19]
Methanol-water (8:2)	Wheat flour	After	Aqueous Buffer	Polymer Dots and AgNPs	FRET	0.3 pg/mL (0.0003 ppb)	-	[45]
Methanol-water (8:2)	Peanut and corn	After	Aqueous Buffer	N,C-dots and AuNPs	FRET	5 pg/mL (0.005 ppb)	-	[76]
Methanol-water (8:2)	Wheat, rice and corn	ND	Aqueous Buffer	AgNCs	Fluorescence	0.3 pg/mL (0.0003 ppb)	-	[50]
Methanol-water (6:4)	Peanuts and rice	ND	Water	AuNPs/CdTeQDs	FRET	3.4 nM (1.06 ppb)	-	[44]
Methanol-water (6:4)	Peanut	Before	Aqueous Buffer	AuNPs/CdZnTe QDs	FRET	20 pg/mL (0.02 ppb)	ND	[77]
Methanol-water (5:5)	Flour	After	Water	UCNPs/Bi_2_S_3_ nanorods	Electrochemical	7.9 pg/mL (0.008 ppb)	-	[48]
Methanol (100%)	Wheat flour	After	Water	Carbon QDs/Cu_2_O NPs	Electrochemical	0.9 ag/mL (9 · 10^−10^ ppb)	-	[70]
Methanol-water (5:5)	Maize meal	After	Water	Ag/Au NPs	SERS	0.03 ng/mL (0.03 ppb)	-	[78]
Acetonitrile pure	11 ingredients	Before	Acetonitrile	AuNPs	Colorimetric	5 μg/kg (ppb)	0.025	This study

ND: Not detailed. Before/after: AFB1 spiking carried out before or after extraction.

## Data Availability

The data presented in this study are available on reasonable request from the corresponding author. The data are not publicly available due to privacy constraints.

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
