# Peer review of "A Novel Preanalytical Strategy Enabling Application of a Colorimetric Nanoaptasensor for On-Site Detection of AFB1 in Cattle Feed"

_sensors, 2022, doi:10.3390/s22239280_

Round 1

Reviewer 1 Report

A short paraghraph explaining why the specific binding NAS/AFB1 induces the aggregation of AuNPs should be inserted.

The part of section 4 Discussion, from row 291 to 317, should be moved to the Introduction section

Reviewer 2 Report

The manuscript is about an optical aptasensor focused on extraction protocols before analytical measurements. It is well organized and written, however, I have some questions and suggestions about your goals in this work and then its statement way in the text. 

1. Please explain the novelty of your work. Your designed aptasensor is very simple and basic. It seems you have focused on the extraction method, not aptasensor design. So you should change the abstract, and introduction to emphasize your work's novelty.

2. The title structure is not suitable structurally, it needs major modification.

3. You should measure the selectivity of your aptasensor in presence of similar molecules. 
